# Transient formation of supramolecular complexes between hyaluronan and oligopeptides at submicromolar concentration

Miguel Riopedre-Fernandez [1,7], Bingxin Chu[2,7], Anna Kuffel [3,4,7], Arianna Marchioro [2 ✉], Denys Biriukov [1,5,6 ✉] & Hector Martinez-Seara [1 ✉]

Charged polymer interactions govern biological and technological processes by altering the structure and dynamics of surrounding water. Studying these interactions across a broad concentration range is challenging, particularly at submicromolar levels where traditional methods lack sensitivity or molecular resolution. Here, we investigate interactions between hyaluronan (HA), a biologically and technologically relevant polymer, and model oligopeptides—nonaarginine, nonalysine, and nonaglycine. By combining angle-resolved second harmonic scattering (AR-SHS), dynamic light scattering, nuclear magnetic resonance, and all-atom molecular dynamics simulations, we resolve the molecular-scale mechanisms and structure of HA–peptide interactions. Our findings reveal selective, multivalent binding between HA and cationic peptides, inducing solvent and solute restructuring and nanoscale clustering. Simulations provide atomic-level insight, elucidating the transient nature of the interactions and highlighting the distinctive behavior of arginine-rich peptides. Our approach, integrating AR-SHS with simulations and routine techniques, offers molecular insights into polymer mixtures and a foundation for future studies of dynamic supramolecular systems in soft materials.

Polymers play a crucial role in crafting water-rich environments, which are essential in a wide range of fields across technology (e.g., water purification systems[1] and biocompatible implants[2]), biology (e.g., glycocalyx[3] and molecular condensates[4]), and pharmaceutics (e.g., drug carriers[5] and hydrogels[6]). All these systems share water as their primary component, while polymers, especially polyelectrolytes, are key in shaping the solution's three-dimensional structure and distinct properties[7]. Despite the molecular simplicity of polymer-rich systems, understanding the factors that control their properties remains challenging[8]. The inherent flexibility of polymers, the structural changes they induce in the surrounding water environment, and the typically dynamic nature of these systems often limit the application of traditional structural techniques. However, gaining insights into the missing molecular details—especially the subtle interactions critical for solute remodeling—is essential for optimizing these systems for the aforementioned applications and advancing our understanding of related biological functions[8].

Hyaluronic acid (HA), or hyaluronan, is a high-molecular-weight, negatively charged polysaccharide that forms polymeric, water-rich media[3]. HA is an excellent example of a polymer with a simple chemical structure that offers many possible applications. Naturally, HA is found in the extracellular matrix (ECM) of various tissues[3], where it plays a pivotal role in cell adhesion and motility[9,10] and contributes to maintaining the mechanical properties of tissues[11,12]. HA's size variability, high solubility in water, viscoelasticity, biocompatibility, and non-immunogenic properties, along with being a suitable polymer for chemical modifications[13], make it widely used for technological, biological, and medical applications[14], including drug delivery[15,16], tissue engineering[17,18], and design of biocompatible hydrogels[19–21].

[1]Institute of Organic Chemistry and Biochemistry of the Czech Academy of Sciences, Prague, Czech Republic. [2]Laboratory for fundamental BioPhotonics (LBP), Institute of Bioengineering (IBI), School of Engineering (STI), Ecole polytechnique fédérale de Lausanne (EPFL), CH-1015 Lausanne, Switzerland. [3]Faculty of Chemistry, Gdańsk University of Technology, Gdańsk, Poland. [4]BioTechMed Center, Gdańsk University of Technology, Gdańsk, Poland. [5]Central European Institute of Technology, Masaryk University, Brno, Czech Republic. [6]National Centre for Biomolecular Research, Faculty of Science, Masaryk University, Brno, Czech Republic. [7]These authors contributed equally: Miguel Riopedre-Fernandez, Bingxin Chu, Anna Kuffel. ✉e-mail: arianna.marchioro@epfl.ch; denysbiriukov@gmail.com; hseara@gmail.com

Interestingly, HA shows gel-like behavior on its own[12], even though HA does not readily self-aggregate or cross-link in water[22,23], mainly due to its negative charge and strong hydration[24]. At the same time, putative intermolecular complexes can form under certain conditions, such as acidic pH[25,26] or the presence of specific mediating molecules[21,27–29]. The latter is often preferred in technical applications over utilizing pure HA hydrogels, which are prone to degradation and have subpar mechanical characteristics[30,31]. However, designing these modified HA-based hydrogels is more challenging, as it requires careful consideration of factors such as the preferable molecular size of both HA and external ligands, operational concentrations, and thermodynamic conditions. All these factors ultimately influence the formation and properties of the resulting supramolecular complexes[32].

The choice and design of HA-binding molecules often take inspiration from known HA–protein interactions, such as those with cell surface receptors and other components of the ECM[33–35]. Since protein HA-binding sites are frequently rich in arginine[36,37], arginine-rich peptides have emerged as promising HA-modulating agents, *e.g.*, in the context of designing drug carriers[38,39]. Beyond biocompatibility, arginine-rich peptides offer high selectivity with HA due to favorable electrostatic and hydrophobic interactions provided by the unique structure of the guanidinium side chain[40]. More broadly, cationic peptides have been widely exploited to form supramolecular complexes with HA yielding polyelectrolyte complexes, multilayer films, core–shell nanostructures, and peptide-templated hydrogels[15,16,21,28,31,32,38,41–50]. For example, polyarginines and HA spontaneously co-assemble into nanoparticles and porous "ionic nanocomplex" sponges and have been used to carry small molecules[38,41,42]. Related platforms utilize lysine-rich polypeptides with HA in several biomaterials (e.g., layer-by-layer films, polyelectrolyte complexes, and thermoresponsive nanogels), which enable storage and controlled release of drugs across physiological ionic strengths[43–46]. Peptide motifs rich in arginines and lysines have been used together with HA in polyplex formulations and HA-decorated micelle nanocarriers[15,47–49,51]. Finally, peptide can co-assemble with HA into supramolecular hydrogels, and HA can template peptide nanofibers into ordered bundles[21,28,50]. These examples have framed our interest in arginine/lysine-rich peptides for remodeling HA solutions.

Recently, combining nuclear magnetic resonance (NMR) spectroscopy with all-atom molecular dynamics (AA-MD) simulations, we demonstrated that tetraarginine peptides indeed interact more strongly with HA octamers compared to equally charged tetralysine peptides[52]. However, the mechanisms by which arginine-rich peptides induce solute remodeling and affect the surrounding aqueous solution remain unclear since longer chains/sequences are typically used in industrial applications or found in biological environments[12]. This knowledge gap—common for other polymers as well—impedes the efficient and controlled design of polymer-rich environments. Moreover, the most commonly used structural techniques are insufficient, especially when used alone, and struggle to provide the essential molecular details, such as the lifetime and relative strength of the underlying interactions. This is also true for HA–peptide interactions, which are characterized by weak affinity, multivalent binding, and generally short-lived interactions[36,53,54]. As a result, there are conflicting views regarding the stability of HA–peptide supramolecular complexes and how it is affected by the molecular size and surrounding conditions[52,55,56]. We thus require new approaches that can provide the size of the formed supramolecular complexes, their three-dimensional arrangement, stability over time, the atomistic patterns of the underlying interactions, their concentration dependence, and the structure of the surrounding hydration shells.

Angle-resolved femtosecond elastic second-harmonic scattering (AR-fs-ESHS, hereafter referred to as AR-SHS) is a powerful technique for monitoring changes in aqueous environments[57]. AR-SHS detects anisotropies in the liquid medium via nonlinear light scattering, effectively probing the orientational order of water molecules[57]. This water ordering is largely influenced by the properties of solutes[58], which makes AR-SHS particularly useful for studying the organization of polymers in a solution, as has been demonstrated in AR-SHS studies on aqueous polyelectrolytes[59],

including pure HA[60]. Water (re-)structuring is anticipated to be exceptionally prominent when solutes form supramolecular complexes, as in HA–peptide systems. Furthermore, the AR-SHS technique operates at submicromolar concentrations[61], *i.e.*, at biologically relevant concentrations for HA in the ECM[12], which are often beyond the reach of conventional techniques.

AR-SHS has some limitations. Interpreting the measured AR-SHS signal is challenging, as it cannot be easily deconvoluted to reveal its molecular origins and requires theoretical models for proper interpretation. This complexity is particularly pronounced in certain soft supramolecular environments, such as HA–peptide solutions, which can appear unstructured at first glance. Fortunately, all-atom molecular dynamics (AA-MD) simulations offer an excellent method for modeling multicomponent solutions and capturing their inherent complexity, including HA–peptide mixtures. Our recent advances in developing reliable models for charged saccharides[52,62,63], which correct for overestimated saccharide–saccharide and saccharide–amino acid interactions[64,65], have further improved this approach. While both AR-SHS and AA-MD have limitations, their combination—especially when coupled with more traditional structural techniques—solves many of the challenges mentioned above and provides a powerful toolkit for investigating polymer-rich systems with solutes of relevant sizes.

This study provides an in-depth molecular view of the structure and interactions of HA–peptide systems. We combine AR-SHS, multi-angle dynamic light scattering (DLS), and NMR experiments alongside large-scale AA-MD simulations. This integrated approach enhances the interpretation of experimental data and enables the collection of a comprehensive molecular picture of HA–peptide interactions, their supramolecular organization, and the specific properties of the resulting aqueous environment. Here, we primarily focus on the unique behavior of arginine-rich peptides, namely nonarginines, and their interactions with large ( >1000 kDa) HA polymers. Bridging multiple conceptually different experimental techniques with realistic computer simulations, we also resolve existing discrepancies regarding the strength and nature of HA–peptide interactions[52,55,56]. In particular, we use AA-MD simulations to interpret highly sensitive AR-SHS signals collected from a binary mixture containing both HA and peptides. For the AA-MD simulations, we employ our newly developed prosECCo75 force field[62], which accurately models interactions between charged species, including saccharides and peptides that are central to this work. Our methodology is thus well-suited for characterizing complex, industrially relevant systems, shedding light on the strength and selectivity of HA–peptide interactions and their subsequent remodeling of the solution environment. In summary, our work presents a detailed and systematic approach for characterizing molecular interactions in polymer-rich systems, with implications for both materials science (e.g., hydrogels for drug delivery or tissue engineering) and biological processes (e.g., molecular condensates).

## Results

Using a combination of NMR and AA-MD simulations, we previously found that HA interacts more strongly with tetraarginine peptides than equally charged tetralysine or charge-neutral and essentially non-binding tetraglycine peptides[52]. This enhanced affinity arises from the partially hydrophobic interactions of arginine side chains with HA, extending beyond simple electrostatic forces. However, these interactions did not lead to stable intermolecular complexes, contradicting earlier reports[55,56]. This discrepancy suggested that the stability of HA–polyarginine complexes may depend on the size of both components[55]. For example, previous studies have suggested that polyarginines longer than octamers may facilitate the formation of stable intermolecular complexes, whereas shorter chains, such as tetramers used in our earlier study[52], might be too short to induce such stability[55].

To explore potential size-dependent effects, we use Nuclear Overhauser effect (NOE-NMR) experiments on significantly larger HA moieties (8–15 KDa, i.e., ~20–40 disaccharides) in the presence of longer polypeptides (nonaarginine R9 and nonalysine K9) at ~20 millimolar

**Fig. 1 | Macroscopic and microscopic view of HA–peptide solutions.** (Top) Chemical structures of arginine (monomer of R9), lysine (K9), glycine (G9), and the disaccharide of HA. The letter "n" indicates a variable number, depending on the experiment. The complete structures can be seen in Supplementary Fig. 1. (Bottom-Left) Microscopy images of HA–R9 and HA–K9 mixtures containing 1375 kDa HA at a 0.87 HA disaccharide/amino acid ratio, taken at ×10 magnification. (Bottom-Right) Unmagnified view of the samples in their containers. Small white arrows point at the solute aggregates. A larger collection of images and videos is available on Zenodo https://doi.org/10.5281/zenodo.15115544.

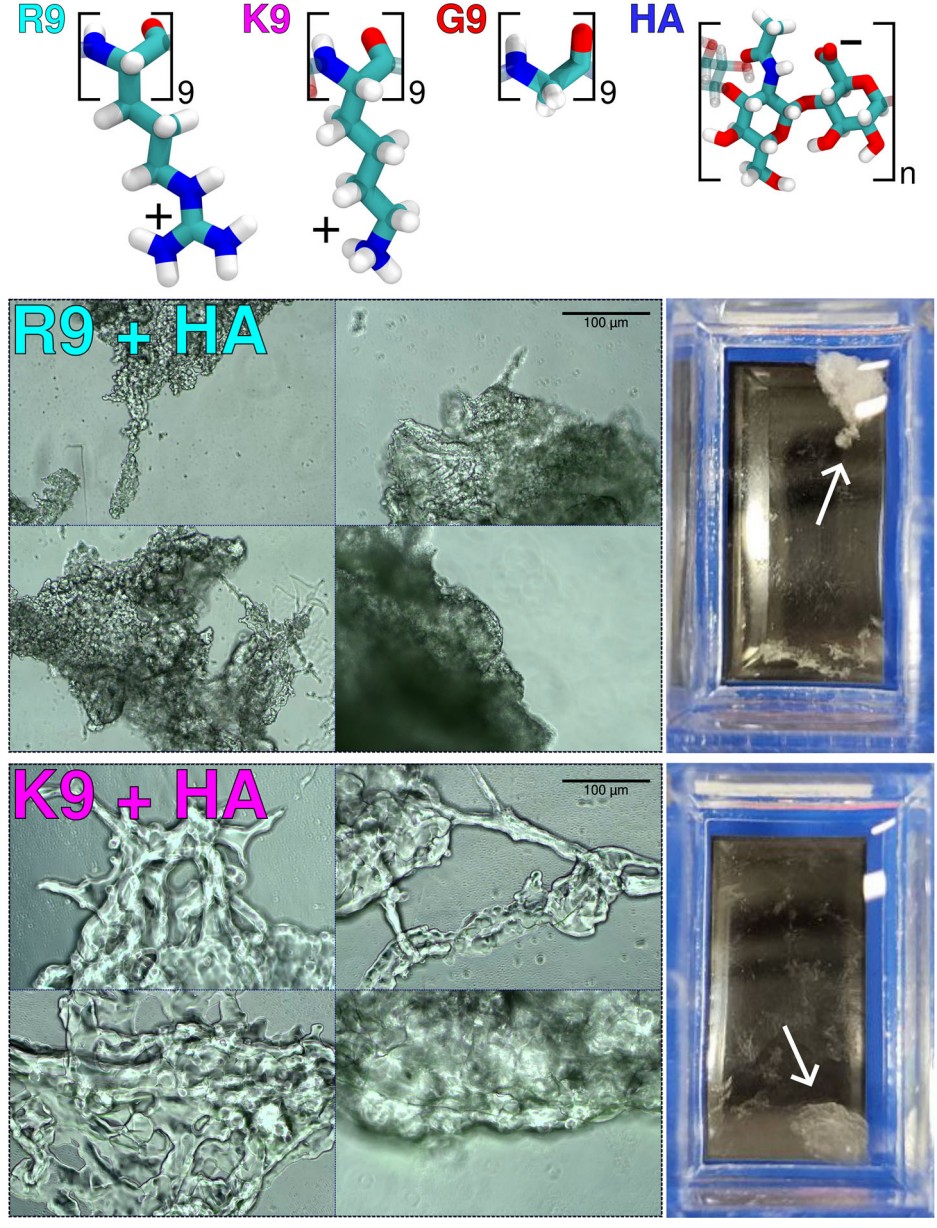

concentrations (see Methods). The resulting solutions appeared clouded immediately after preparation but became homogeneous and clear shortly after, allowing for the collection of their NOESY spectra, see Supplementary Figs. 2, 3. Despite using solutions at high millimolar concentrations—well above the typical nanomolar to micromolar range found in biological systems—these larger molecules remained fully soluble upon mixing. We observed no intermolecular NOESY signals, consistent with our previous results from smaller molecular systems[52]. This result alone suggests that previous views, driven mainly by ill-posed computational techniques, overestimate the HA interaction with cationic peptides, providing an unrealistic static picture of this dynamic binding.

It is now interesting to explore the behavior of even larger HA molecules, specifically in the industrially and biologically relevant megadalton size range (1250–1500 kDa, i.e., ~3125–3750 disaccharides, hereafter referred to as 1375 kDa for simplicity). Unlike smaller 8–15 kDa HA, larger HA induces precipitation in HA–R9 and HA–K9 solutions, despite similar HA mass concentrations and even lower peptide-to-HA ratios, see Fig. 1. Notably, the heterogeneous solutions of HA–R9 and HA–K9 are visibly distinct. HA–R9 mixture forms a more compact and opaque precipitate,

whereas adding K9 to HA results in more dispersed filaments and sheets throughout the cuvette. Additionally, in HA–R9 mixtures, most of the material collapses into a single, well-defined blob, while in HA–K9 mixtures, more fragmented structures appear. The edges of the HA–R9 precipitate are sharper, and its extra material exhibits a different morphology compared to HA–K9. In contrast, the reference HA–G9 mixtures remain slightly translucent, similar to pure G9, and exhibit much lower macroscopic heterogeneities—even at higher concentrations—unlike the pronounced changes observed with R9 and K9 (see Supplementary Fig. 6). The heterogenities in the HA–peptide mixtures disappeared by adding NaCl concentration (i.e., 0.2 M or 0.5 M NaCl). The disappearance of the aggregates in a salt-dependent manner (Supplementary Fig. 6) indicates that the phenomenon has an electrostatic origin. Given that the only charged groups in G9 are the peptide termini, they must be responsible for the small observed aggregation. The termini are also charged in R9 and K9, but the additional positive charges in the side chains in large part explain the stronger aggregation at low ionic strength.

The heterogeneity of the studied mixtures prevents direct NMR measurements, and lowering the concentration to improve homogeneity

**A** 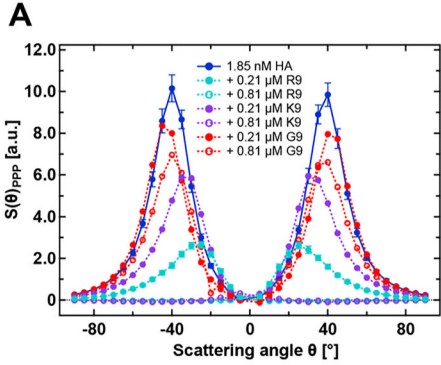

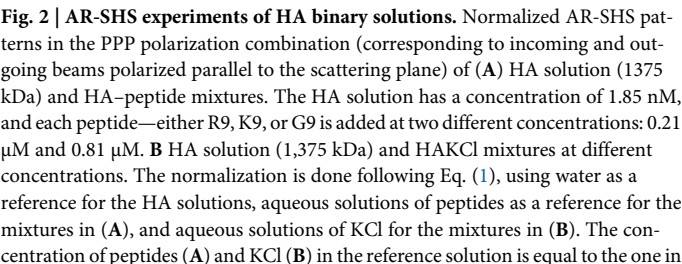

**B** 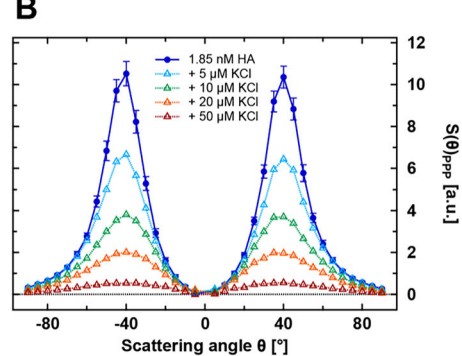

**Fig. 2 | AR-SHS experiments of HA binary solutions.** Normalized AR-SHS patterns in the PPP polarization combination (corresponding to incoming and outgoing beams polarized parallel to the scattering plane) of (**A**) HA solution (1375 kDa) and HA–peptide mixtures. The HA solution has a concentration of 1.85 nM, and each peptide—either R9, K9, or G9 is added at two different concentrations: 0.21 µM and 0.81 µM. **B** HA solution (1,375 kDa) and HAKCl mixtures at different concentrations. The normalization is done following Eq. (1), using water as a reference for the HA solutions, aqueous solutions of peptides as a reference for the mixtures in (**A**), and aqueous solutions of KCl for the mixtures in (**B**). The concentration of peptides (**A**) and KCl (**B**) in the reference solution is equal to the one in

the corresponding mixture. The error bars are shown only for pure HA and HA–R9 mixtures for clarity; error bars for the other mixtures are similar to those of HA–R9. These error bars represent the error propagation calculated for the normalized patterns, based on the standard deviation of 20 measurements. Panel **A** shows that all peptides modify substantially the shape and intensity of the corresponding pattern, with R9 and K9 having the strongest effect, especially at 0.81 µM. Panel **B** shows that the effect of KCl is less pronounced than in the presence of R9 or K9. The AR-SHS patterns for pure peptide solutions (Supplementary Fig. 7), for the mixtures in the PSS polarization combination (Supplementary Fig. 8), as well as data for HA–R9 mixtures at additional concentration ratios (Supplementary Fig. 9) are also provided.

would push the system below the NMR detection threshold. Although we can record $^1$H spectra of the supernatants, this only provides a limited perspective (see Supplementary Figs. 4, 5 and Supplementary Note 1). Therefore, reliance on NMR and similar conventional techniques, which require relatively high concentrations, restricts our ability to investigate the behavior of large HA and its complexation processes.

Instead, using AR-SHS allows us to study high-molecular-weight 1,375 kDA HA at nanomolar concentration mixed with R9, K9, and G9 peptides at submicromolar concentrations. Briefly, in AR-SHS, two photons of a powerful laser beam induce a second-order polarization of the molecules present in the solution, which leads to the emission of one photon of doubled frequency. Photons are collected at 180° around the sample, yielding an angle-resolved scattering pattern. For nonresonant excitation of a solute/solvent system, the main contribution to the second harmonic (SH) emission originates from the solvent, in our case, water. Indeed, the SH emission is sensitive to the orientational order of solvent molecules, also referred to as orientational correlations. These can be induced by either molecule-specific chemical interactions between the solute and solvent, such as modifications to the hydrogen bond network or the remodeling of solute complexes within the solution, or by electrostatic-field interactions, where the electrostatic field generated between a charged solute and surrounding electrolyte ions can align water dipoles in the solution[57]. The extent of the latter electrostatic effect depends on the ionic strength of the solution; a higher ionic strength leads to a more effective screening of the electrostatic field. Overall, the intensity of the AR-SHS signal in solute/solvent systems reflects the solute's capability to orient the surrounding solvent molecules. Orientational order that is on average spherically symmetric appears in the PPP and PSS polarization combinations (see Methods for definitions and further details)[66]. Practically, PPP yields the highest signal intensity and is thus the preferred metric to report. The raw AR-SHS data are processed by first subtracting the signal intensity of the corresponding reference solution from that of the sample, and subsequently normalized to pure water to provide comparability (see Eq. (1) in Methods). Consequently, a higher AR-SHS intensity of these normalized patterns (denoted by S(θ)$_{PPP}$) indicates a higher degree of orientational order (orientational correlations) relative to the corresponding reference solution.

The normalized AR-SHS patterns in the PPP polarization combination for pure HA and HA–peptide mixtures are presented in Fig. 2A. It shows that HA alone displays a strong AR-SHS signal at the selected concentration, consistent with the previous AR-SHS study on pure HA[60]. For HA, the

reference is pure water, which ensures that the AR-SHS patterns report on the orientational order of water induced by HA. The strong collected signal in HA can be attributed to its significant concentration of carboxyl groups—one per disaccharide[60]. The accumulation of a high density of electric charges along the polymer chain is believed to produce a significantly stronger localized electrostatic field than randomly distributed free ions, resulting in a more pronounced effect on the hydrogen-bond network of water and enhanced water–water correlations[60].

Now, we focus on the pattern features specific to the HA–peptide mixtures. Two peptide concentrations were tested for each mixture. The first one, referred to as "excess", contains an excess of HA, namely 1.85 nM HA and 0.21 µM peptide, corresponding to a disaccharide-to-amino acid molar ratio of 3.33:1. The second one, referred to as "equimolar", features approximately equal amounts of HA disaccharides and peptide amino acids and an approximate charge balance (1.85 nM HA and 0.81 µM peptide, yielding a molar HA dimer-to-amino acid ratio of 0.87:1).

Our results unambiguously demonstrate a distinct effect on the AR-SHS signal upon adding R9, K9, or G9 peptides. At the HA excess ratio, R9 peptides cause a significant decrease and inward shift of the signal peaks. K9 peptides produce a similar but less pronounced effect, while G9 peptides only mildly reduce the intensity of the AR-SHS peaks without any shifts. At the higher peptide concentration (equimolar ratio), both R9 and K9 peptides fully attenuate the signal intensity, occasionally producing even rare small negative values, see Supplementary Fig. 9. Conversely, increasing G9 peptide concentration only slightly decreases the signal amplitude.

The observed decrease in AR-SHS signal intensity upon peptide addition indicates a reduction in the orientational ordering of water around HA, compared to the pure HA solution. Here, the contribution from the corresponding peptide reference solution—containing the same peptide concentration in pure water—has already been subtracted from the data in Fig. 2A, ensuring that the observed changes primarily arise from HA–peptide interactions. Note that the peptides alone exert a relatively minor influence on the water orientational order, with the AR-SHS signal intensity being approximately an order of magnitude smaller than that for the pure HA solution, see Supplementary Fig. 9.

Overall, it is evident that both positively charged peptides (R9 and K9) have a stronger effect on the AR-SHS pattern when combined with HA than the charge-neutral G9, with R9 having a more significant effect. However, the observed AR-SHS signal attenuation alone does not conclusively

determine whether the reduced orientational order of water molecules following peptide addition is caused by changes in solute–solvent interactions—such as a decrease in water-accessible HA surface area—or by alterations in electrostatic-field interactions due to the increased ionic strength. Because the peptides are charged, their presence could lead to a higher degree of electrostatic screening within the solution, effectively reducing the influence of HA on water–water correlations (the conductivity and pH of the peptide solutions are summarized in Supplementary Table 1 and Supplementary Note 2 for completeness).

To distinguish between these two possible contributions, we examined the effect of adding increasing amounts of KCl to HA solutions, Fig. 2B. KCl, because of its low charge density, is expected not to interact substantially with HA. The collected data show that as KCl concentration increases, the AR-SHS signal intensity decreases without shifting the position of the maxima, qualitatively mirroring the effect of adding G9. The attenuation is also less pronounced with KCl, signaling a larger effect of R9 and K9. Note that the increase of ionic strength resulting from adding 0.21 µM or 0.81 µM of R9/K9 to HA solutions is equivalent to adding approximately ~2 µM or ~8 µM KCl, respectively. Furthermore, unlike in the HA–R9 and HA–K9 mixtures, no complete signal attenuation is observed even at 50 µM KCl.

Since signal attenuation is beyond the expected ionic-screening effect induced by the addition of charged compounds, R9 and K9 peptides are also likely to reduce the amount of HA that can interact freely with water (see Supplementary Fig. 10 showing the change in solvent-accessible surface area, and also discussed below), indicating a significant contact between the peptides and HA even at low concentrations. Importantly, the fact that adding KCl does not shift the position of the maximum intensity confirms that electrostatic screening alone cannot account for the observed changes in the AR-SHS signals in the presence of R9 and K9, where the peak of the signal shifts toward smaller (forward) scattering angles. Previous studies on solid particles have shown that the inward shift of AR-SHS signal intensity is attributed to the presence of larger-sized particles that favor forward scattering[67,68]. Should aggregates form upon the addition of peptides to HA, the same effect is expected to apply to our AR-SHS data. We can, therefore, hypothesize that the AR-SHS data in Fig. 2A indicates the formation of aggregates upon the addition of R9 and K9.

To further verify that the changes and, in particular, angular shifts in our AR-SHS patterns can also be attributed to the formation of aggregates (or rather detectable intermolecular complexes since the solutions at submicromolar concentration are visually fully homogeneous), we conducted multi-angle dynamic light scattering (DLS) measurements on the same samples used for AR-SHS (see "Methods"). DLS is typically used to measure the hydrodynamic diameter, which represents the size of hypothetical, smooth, spherical particles that diffuse at the same rate as the sample particles. In our DLS setup, three angles are measured to improve the detection of several size-distinct populations. For complex samples with arbitrary geometry, DLS does not directly yield quantitative information about the size, but it can indicate whether a scattering supramolecular structure is formed in solution. Furthermore, the polydispersity index (PI) can give an indication of the broadness of the size distribution, with values above 0.7 representing very broad size distributions.

The DLS results indeed indicate significant differences in supramolecular cluster formation among the studied HA–peptide mixtures, see Table 1. No well-defined structures were detected through DLS in pure HA solutions or when adding just KCl; for these samples, at least one of the three DLS angles could not be measured, and the remaining angles showed one or multiple populations with a size above 1000 nm (data marked by a *). Upon addition of R9, a very weak light scattering signal is already observed for peptide concentrations as low as ~0.05 µM, while still allowing for light collection at three different angles (data marked by a **). For R9 concentrations starting at ~0.1 µM, DLS can clearly identify structures with sizes ranging from approximately 100–150 nm. In contrast, adding K9 peptides leads to distinctly measurable submicrometer structures only at higher peptide concentrations (~0.2 µM), and the observed structures exhibit greater size variability (large PI even at high concentrations). Finally, G9

peptides do not induce the formation of any well-defined submicrometer structures, even at the highest concentration, similar to the case of pure HA solutions. These findings indicate that the formation of distinct, uncharacterized supramolecular structures between HA and different peptides varies significantly, with R9 having the largest impact. This variation directly explains and correlates with the observed changes in the AR-SHS scattering patterns.

All three experimental methods—NMR, AR-SHS, and DLS—consistently show a clear interplay between HA and R9/K9 peptides, in contrast to the minimal interaction observed with G9. Still, neither technique can provide an atomistic view of the underlying structures and interactions responsible for such observations. To address this, we employed AA-MD simulations on systems designed to replicate the conditions of the experimental measurements as closely as possible, see Methods for details. In brief, aqueous solutions containing five HA octadecamers were simulated alongside five R9, K9, or G9 peptides, closely reproducing the experimental equimolar ratio. Accessible length and time scales in AA-MD simulations restricted us to concentrations of HA and peptides around 6 mM.

First of all, AA-MD simulations indicate a substantial reduction in the water-accessible surface area for HA in the presence of R9, a notable but smaller decrease with K9, and virtually no change with G9, see Supplementary Fig. 10. Further analysis reveals distinct cluster formation tendencies depending on the peptides present, Fig. 3. HA–G9 mixtures show minimal aggregate formation, and the system behavior is visually (Fig. 3A) and qualitatively (Fig. 3B) similar to pure aqueous HA. The most common cluster in HA–G9 mixtures consists of 2–3 solute molecules. In contrast, adding K9 induces more extensive aggregation, resulting in a variety of larger clusters. The probability of having the largest aggregates composed of ten solute molecules is ~20%. Intermediate sizes, where the largest aggregate is of 4–9 units, have a similar probability. Finally, adding R9 peptides leads to the most notable aggregation, mostly resulting in a single cluster containing all available solutes. Still, interactions between individual R9 peptides and HA chains remain transient (Supplementary Fig. 11). The characteristic decay time of $HA_i$–$HA_j$ distances, which correlates with the molecular diffusion within the cluster, remains fast near ~50 ns in HA–R9 solutions, despite being comparatively longer than in HA–K9 and HA–G9 mixtures (Fig. 3C). The same is true for $HA_i$–$peptide_j$ distances and contacts (Supplementary Figs. 12, 13). Thus, HA–R9 clusters behave as dynamic supramolecular complexes characterized by constant changes in constituent molecules and multivalent interactions. This multivalency allows relatively weak interactions to hold the cluster together—when one interaction breaks, a new one quickly forms (Supplementary Fig. 11). The multivalency also serves as an explanation for the increased binding of larger polymers, where multiple simultaneous interactions are more probable. Moreover, the planar guanidinium side chain of R9 allows it to engage in non-polar interactions with the HA chains, in contrast with the purely electrostatic interaction of K9[52]. This additional non-polar interaction directly contributes to the higher aggregation and slower dynamics of HA–R9 mixtures. Overall, all these AA-MD observations and trends perfectly align with the interpretation of our NMR, AR-SHS, and DLS measurements and agree with the previously reported behavior of HA solutions with short tetrapeptides[52].

Next, to better understand how solutes and water molecules may contribute to the AR-SHS signal, we analyze the orientation of solute molecules in the HA–peptide mixtures, Fig. 3D. The angular distribution of HA molecules with respect to each other in a pure HA solution indicates no preferential orientation of HA polymers. When G9 or K9 are added, the average orientation of the HA chains relative to each other remains mostly unaffected. However, the addition of R9 peptides significantly influences it, prompting HA molecules to preferentially align in parallel (or antiparallel) orientations. This preference can be seen from the high probability of small angles, indicating (anti)parallel alignment of the molecules. In contrast, angles near 90°, which reflect perpendicular orientations, are underrepresented in mixtures with R9. Interestingly, R9 peptides themselves also display a preference for being aligned parallel to HA chains. These angular distributions are a consequence of the formation of transient structures

**Table 1 | Measured size (see Methods for details), polydispersity index (PI), and standard deviation (STD) from multi-angle dynamic light scattering (DLS) measurements of an HA solution (1375 kDa) and HA–peptide mixtures**

| Sample | DLS size [nm] | PI | STD [nm] |
|---|---|---|---|
| HA 1.85 nM | > 1000[*] | 0.96 | – |
| HA 1.85 nM + KCl 5–50 µM | > 1000[*] | – | – |
| HA 1.85 nM + R9 0.05 µM | 174[**] | 0.35 | – |
| HA 1.85 nM + R9 0.10 µM | 102 | 0.19 | 49 |
| HA 1.85 nM + R9 0.21 µM | 97 | 0.24 | 65 |
| **HA 1.85 nM + R9 0.26 µM** | **117** | **0.10** | **44** |
| **HA 1.85 nM + R9 0.30 µM** | **150** | **0.02** | **34** |
| **HA 1.85 nM + R9 0.41 µM** | **147** | **0.01** | **42** |
| **HA 1.85 nM + R9 0.81 µM** | **113** | **0.03** | **29** |
| HA 1.85 nM + K9 0.10 µM | 330[**] | 0.17 | – |
| HA 1.85 nM + K9 0.21 µM | 97 | 0.24 | 50 |
| HA 1.85 nM + K9 0.26 µM | 54 | 0.23 | 29 |
| HA 1.85 nM + K9 0.41 µM | 65 | 0.22 | 21 |
| HA 1.85 nM + K9 0.81 µM | 390 | 0.12 | 104 |
| HA 1.85 nM + G9 0.21 µM | >1000[*] | 0.7 | – |
| HA 1.85 nM + G9 0.30 µM | >1000[*] | 0.8 | – |
| HA 1.85 nM + G9 0.81 µM | >1000[*] | 1 | – |

The measured size corresponds to the effective size of the extended hydration shell of solutes, but should only be treated as a qualitative indication, as quantitative analysis is only possible for spherical particles (see the main text and Methods). PI reflects the broadness of the size distribution, where values ≤0.1 denote a narrow distribution (those data are given in bold), and values ≥0.7 suggest a very broad distribution. Data marked with [*] indicate cases where DLS could not measure data at at least one of the three angles, resulting in failed measurements and a high PI. Data marked with [**] indicate a very weak scattering signal.

where HA and R9 molecules associate to form bundles, as shown in Fig. 3A. The formation of these bundles in HA–R9 mixture is consistently observed across simulation replicas and can be characterized by dips in the time-dependent analysis of HA–HA or HA–R9 angles, see Supplementary Fig. 14. These dips are not present in the cases of K9 or G9, Supplementary Fig. 15. The formed HA–R9 bundles might be responsible for the decreased transparency of HA–R9 samples in Fig. 1, although further evidence is needed to confirm this hypothesis. Overall, our findings suggest that solute remodeling occurs in the studied solutions in a peptide-dependent manner, further explaining the observed changes in the AR-SHS signals.

Then, we examined how water orientation changes around HA and peptides, which is likely the main contributor to the observed variations in the AR-SHS signals. Given the distinct structures formed in each mixture, we sought to determine whether the surrounding water structure is also affected. To begin, we calculated the orientation of water molecules in solutions containing either pure HA or pure peptides (see Supplementary Note 3). As shown in Fig. 4A, pure HA solutions induce a noticeable orientation of water molecules, with their hydrogen atoms consistently aligning toward the HA surface (mean angle < 90°). Such water orientation visibly extends up to ~15 Å from HA—and even beyond—aligning with previous predictions[60] and likely being responsible for the AR-SHS signal measured in pure HA solutions. Contrary, positively charged peptides (R9 and K9) induce the reversed water orientation, i.e., water oxygens are oriented toward the peptide surface (mean angle > 90°), Fig. 4A. This effect again extends up to ~15 Å away from the peptides and, interestingly, is almost identical for R9 and K9 solutions. G9 displays an intermediate behavior, with the water mean angle oscillating around 90°, as expected for a neutrally charged peptide. Note that randomly oriented water molecules result in an average 90° angle.

The formation of HA–peptide aggregates significantly affects the water orientation around the solutes. Figure 4B, C shows how the mean water orientation changes as a function of the distance of each water molecule to the closest atom of HA or the peptide in the mixture. The difference in orientation is calculated by subtracting the water orientation in either pure HA (Fig. 4B) or pure peptide (Fig. 4C) solutions of matching concentration

from the water orientation in the mixtures. The subtraction of the single-solute signal from the mixture signal mirrors the approach used in reporting AR-SHS experiments. The HA–G9 mixture does not significantly change water orientation compared to either pure HA or pure G9 solutions. This result aligns with earlier observations of minimal structural remodeling in the HA–G9 systems, as shown in Fig. 3 and also interpreted from the AR-SHS data in Fig. 2. Small changes are also observed when water is simultaneously in contact with both HA and G9, which is expected given the unique peculiarities of this environment. However, only a few water molecules are involved in this kind of interaction.

In contrast, the addition of K9 and especially R9 induces pronounced changes in water orientation even at distances far from the HA and/or peptides (Fig. 4B, C). In the case of Fig. 4B, there is a global increase (shown in red) in the average HA–water angles in the HA–R9 and HA–K9 mixtures that implies a significantly reduced tendency for water molecules to orient their hydrogen atoms toward HA compared to the pure HA solution. At the same time, the negative values in Fig. 4C (shown in blue) indicate that water molecules are less likely to orient their oxygen atoms toward the peptides when HA is present. It is important to note that, although the results in Fig. 4B, C may appear to suggest opposite trends, they, in fact, convey the same underlying message: the orientational order of water is less enforced in HA–R9 and HA–K9 mixtures with respect to the pure HA and peptides solutions. These findings, together with the analyses of water-accessible surface area (Supplementary Fig. 10) and intermolecular complex formation (Fig. 3), effectively explain the decrease in the AR-SHS signal caused by R9 and K9 and support the observation that R9 has a stronger effect: the formation of supramolecular complexes limits the water-accessible area of the solutes, and these complexes possess a lower intrinsic capacity to reorient water compared to the pure solutes.

## Discussion

Polymer-rich aqueous solutions are ubiquitous in biological and industrial contexts. However, the molecular-level processes governing their behavior, such as how solutes interact, reorganize, and influence water structure, remain not fully understood. Here, we combined all-atom molecular

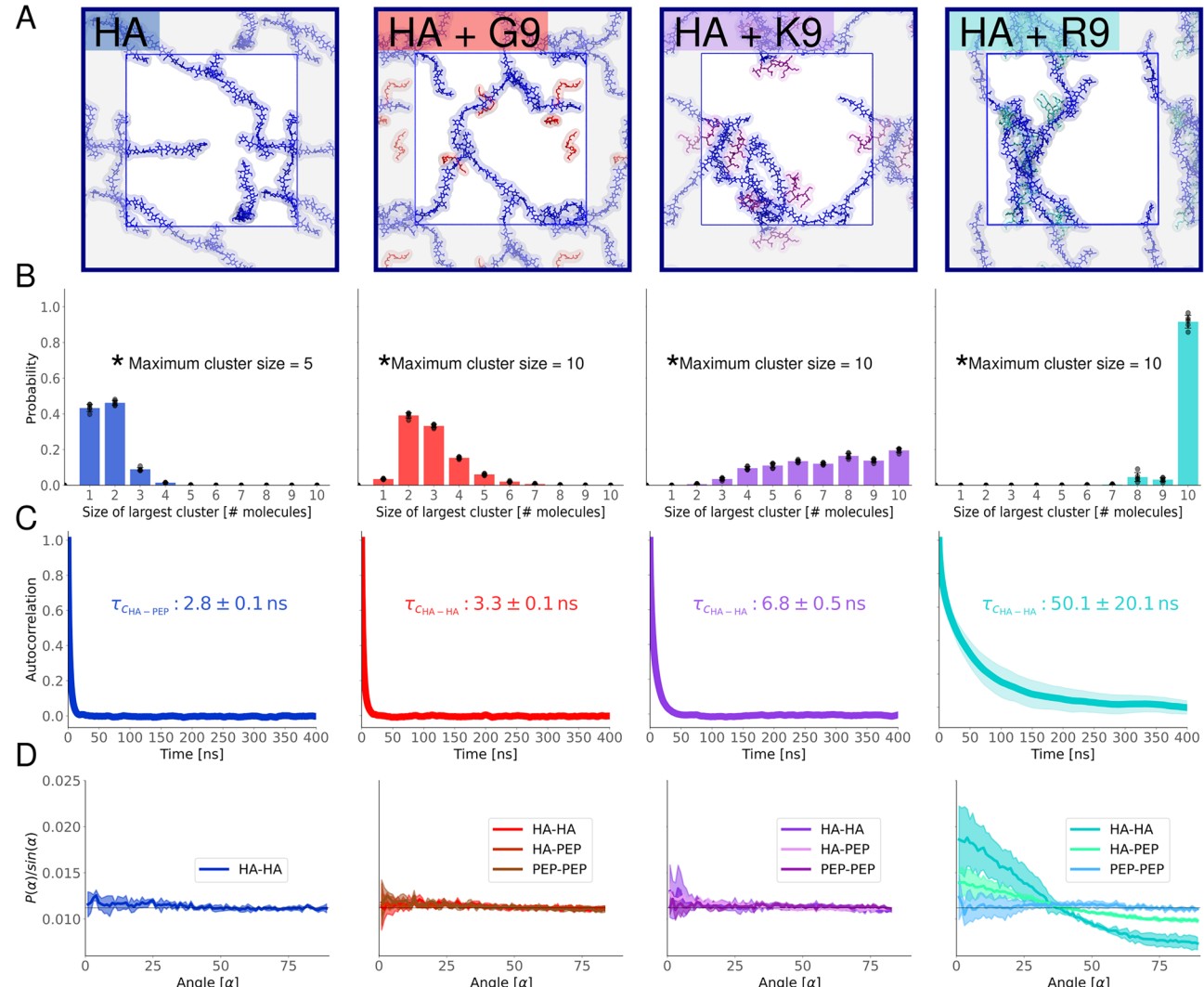

**Fig. 3 | Molecular dynamics simulations of HA–peptide mixtures.**
**A** Representative snapshots showing pure aqueous HA (blue) and its mixtures with peptides: G9 (red), K9 (purple), and R9 (turquoise). **B** Distribution of the largest solute aggregate in each frame. Aggregates are defined based on continuous spatial contact within a 3.5 Å distance cutoff of any solute. The standard deviation between replicates is shown with black bars ($n = 5$), and the individual averages for each replicate are shown as points. **C** Autocorrelation function of HA–HA distances. The minimum distance between every residue pair was included in the time series. Characteristic times were determined by fitting to an exponential decay, $e^{-t/\tau_c}$, with $\tau_c$ representing the

characteristic time. The standard deviation between the five replicate means is shown as shaded areas, but only visible for HA–R9 systems. **D** Angle distributions between molecular pairs categorized as HA–HA (hyaluronan–hyaluronan), HA–PEP (hyaluronan–peptide), and PEP–PEP (peptide–peptide). Angles were determined from molecular axes derived via singular value decomposition. Probability distributions were normalized by the sine of the angle to account for spherical volume and scaled such that the total area under each curve equals one. Standard deviations between replicate means are represented by shaded areas ($n = 5$).

dynamics (AA-MD), angle-resolved second-harmonic scattering (AR-SHS), nuclear magnetic resonance (NMR) spectroscopy, and multi-angle dynamic light scattering (DLS) to achieve a clear molecular understanding of solute remodeling caused by interactions between high-molecular-weight hyaluronan (HA) and oligopeptides in water. Nonaarginine (R9), non-alysine (K9), and nonaglycine (G9) served as relatively simple model peptides to probe different binding behaviors with HA: R9 represents arginine-rich motifs in HA-binding proteins, K9 acts as a lysine-rich counterpart to assess the role of side-chain positive charges, and G9 serves as a neutral control.

Our data reveal that introducing cationic peptides into HA solutions dramatically alters their aqueous environment. AR-SHS experiments demonstrate significant changes in water structure upon peptide addition, indicating substantial reordering of hydration shells due to HA–peptide interactions. This effect is most pronounced for R9, but also notable for K9. DLS further confirms supramolecular complex formation, suggesting the

presence of aggregates in HA–R9 mixtures and, to a lesser extent, in HA–K9. Conversely, no aggregates form with HA alone and, to a large extent, with HA–G9 mixtures. Critically, these phenomena appear even at sub-micromolar concentrations—a regime inaccessible to many conventional structural techniques, underscoring the value of AR-SHS for probing polysaccharide–peptide interactions. These low-concentration conditions closely mimic physiological environments, *e.g.*, extracellular matrices (ECMs) abundant in long HA chains[12], thus highlighting the biological relevance of our findings. However, the macroscopic picture changes at higher millimolar concentrations, where peptide mixtures with high molecular weight HA aggregate significantly, precluding the use of certain structural techniques. Recognizing this concentration-dependent behavior is critical for real-world translation, as it suggests that biological systems or engineered materials could regulate HA–peptide interactions by locally tuning polymer size or concentration to maintain fluidity or induce aggregation.

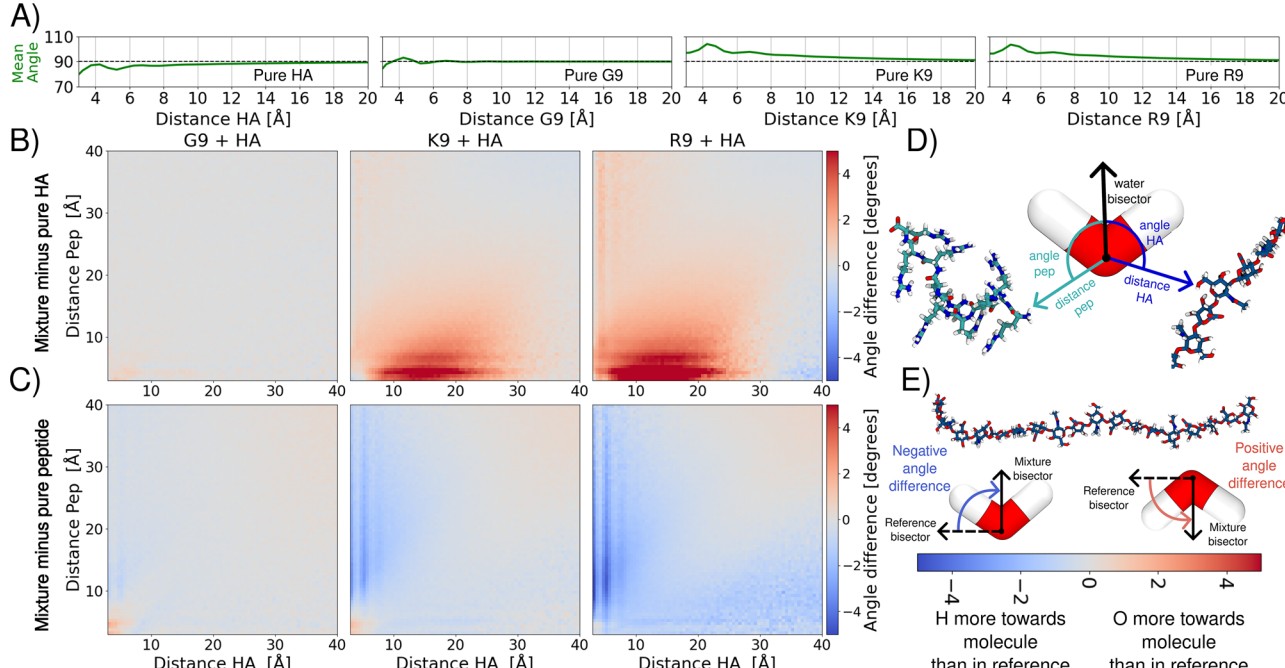

**Fig. 4 | Water orientation in pure and binary HA/peptide solutions. A** Average absolute water angles in the reference systems used for normalization. A 2D version of these figures, along with the absolute average water angles for the mixtures, can be found in Supplementary Figs. 16, 17. Note that water orientation around pure R9 and K9 is almost identical. A similar graph showing the number of water molecules in each system is shown in Supplementary Fig. 18. **B** Average water angles relative to HA molecules, shown as a function of the distance to both HA and peptides. Data are normalized to a system containing only HA. **C** Average water angles relative to the peptides, shown as a function of the distance to both HA and peptides. Data are normalized to a system containing only the respective peptide. **D** Graphical illustration of the metric. For each water molecule, its bisector was determined. The angle between this bisector and the vector pointing from the water's oxygen atom to the nearest atom of either an HA or peptide molecule was then calculated. **E** Cartoon explanation of the color scheme. For panels **A**, **B**, and **C**, the data contain the average of all applicable replicates.

AA-MD simulations enrich our interpretation by providing molecular details. These simulations capture the significant water-structure perturbations in HA–R9 solutions, explained by HA and R9 forming multivalent assemblies that are sustained by multiple transient binding contacts. These contacts continuously break and reform, indicating that the complexes are highly dynamic, rather than static aggregates. Consistent with this interpretation, nuclear Overhauser effect NMR (NOE-NMR) experiments (conducted with medium-sized HA to avoid excessive aggregation) reveal no intermolecular signals in neither HA–R9 and HA–K9 mixtures, indicating the lack of stable, long-lived binding modes. Our simulations further show that HA–peptide aggregates form and dissipate within hundreds of nanoseconds, agreeing well with previous observations on smaller HA–peptide systems[50,52]. Our findings underscore a key feature of the HA–peptide systems: weak, multivalent interactions, especially involving arginine-rich sequences, drive transient clustering of HA chains that perturb water structure, yielding reversible multiscale aggregation. The multivalency also explains the stronger binding between longer HA and peptide polymers. While observed in this specific system, such behavior is likely not unique to it.

The ability of arginine-rich peptides to cluster HA holds significant implications for ECM modulation and therapeutic design. HA, a key component of the ECM, regulates tissue hydration, signaling, and barrier functions. Our results show that peptides like R9 can reorganize HA-rich matrices by locally drawing multiple HA chains together, creating temporary cavities and structural rearrangements. This aligns with the known behavior of many cell-penetrating and ECM-binding peptides: polycationic sequences bind and cross-link polysaccharides on cell surfaces, promoting clustering and facilitating endocytosis[51]. Arginine-rich motifs are also established cell-penetrating sequences[40,69]. Thus, the dual capacity of R9 to interact strongly with HA and to transport cargo across cell membranes presents an intriguing opportunity for targeted drug delivery strategies. More broadly, our insights into HA–peptide interactions and supramolecular aggregation could

inform the design of novel hydrogels and ECM-mimetic materials. Our findings also rationalize why cationic peptide–HA complexes are so versatile in materials and drug delivery. In drug-carrier formats, polyarginine–HA polyelectrolyte complexes can reproducibly form ≈ 100–200 nm objects and can be processed into porous sponges[38,41,42]. Likewise, HA–polylysine architectures stabilize cargos and tune release via ionic strength and cross-link density[43–46]. Finally, prior studies show that HA–peptide mixtures can form cohesive hydrogels stabilized by multiple weak, reversible interactions[21,28,50]. All those designs align with our AR-SHS/DLS/MD evidence that the HA–peptide assemblies are transient even at submicromolar concentrations.

Methodologically, our study highlights the impact of integrating advanced spectroscopic and scattering techniques with computational simulations to unravel complex behaviors in multicomponent soft matter systems. AR-SHS proved to be exceptionally sensitive to molecular organization, detecting subtle changes in water orientation[57], making it ideal for our HA–peptide mixtures that traditional methods cannot resolve. However, interpreting AR-SHS signals from mixed polymer–peptide solutions is challenging. Coupling AR-SHS with DLS, NMR, and AA-MD simulations allowed us to effectively understand the contributions of solute assembly and water reorientation to the measured signals. Simulations were indispensable for assigning experimental observations to precise molecular events, such as HA chain reorganization and alignment, or altered water orientation around interacting molecules. To our knowledge, this work represents the first application of AR-SHS to probe binary polyelectrolyte–peptide interactions; previous AR-SHS studies primarily focused on pure polymers, colloids, or interfacial systems. Successfully deploying AR-SHS on polymer–biomolecule mixtures expands nonlinear optical applications within biophysical chemistry. More generally, our integrated approach offers a template for studying heterogeneous systems, correlating multiple experimental signals, and using molecular simulations to provide an atomistic view in order to achieve a detailed understanding beyond single-method studies.

## Methods

### Chemicals

Sodium hyaluronate (HA, molecular weight 8–15 and 1250–1500 kDa, with the latter referred to as 1375 kDa HA for simplicity) was purchased from Contipro a.s and used as received. Potassium chloride (KCl, > 99.999% trace metals basis, Thermo Fisher Scientific) was used as received. Sodium chloride was also used as received (NaCl, > 99.5%, Sigma-Aldrich). Peptides were synthesized via solid-phase peptide synthesis on Liberty Blue peptide synthesizer (CEM, USA), following standard Fmoc chemistry protocols, DIC/Oxyma coupling reagents, and 2-chlorotritylchloride resin support (0.1 mmol scale, with a double coupling of sevenfold excess of amino acid). All the peptides were uncapped, with a free amine and a free carboxylate at the N- and C-termini, respectively. R9 and K9, bearing side-chain protecting groups, were cleaved off the resin with a mixture of acetic acid/2,2,2-trifluoroethanol/dichloromethane (1:1:3) for 2 hours at room temperature. The resin was filtered, and the combined filtrates were evaporated to dryness. The resulting peptides were then deprotected with a mixture of trifluoroacetic acid/triisopropylsilane/water (95:2.5:2.5) for 1 hour at room temperature. G9 was cleaved directly from the resin using the same trifluoroacetic acid/triisopropylsilane/water mixture (95:2.5:2.5) for 1 hour at room temperature.

The crude peptides were dissolved in water (R9, K9) or trifluoroacetic acid (G9) and injected onto a Maisch C18 reverse phase column Reprosil Gold 120. The solvent flow rate was 5 ml/min. The elution was performed with an initial 20 min wash using water and 0.05% trifluoroacetic acid, followed by a gradient of 1.25% methanol per minute. The HPLC traces are shown in Supplementary Fig. 19. Peptide purity was assessed by analytical reverse-phase high-performance liquid chromatography (Vydac 218TP54 column) and liquid chromatography/mass spectrometry (Agilent Technologies 6230 ToF LC/MS). The MS trace is given in Supplementary Fig. 20.

### Angle-resolved second harmonic scattering

**Sample preparation.** All procedures described hereafter used ultrapure water (Milli-Q, Millipore, Inc., electrical resistance of 18.2 MΩ × cm). For HA and HA–peptide mixture samples, 100 mg of 1375 kDa HA powder was first dissolved in 50 mL of ultrapure water. The stock solution was heated to 40 °C for 3 h, allowed to stand overnight to ensure homogeneity, and vortexed for 1 min prior to usage. The stock solution was subsequently diluted with ultrapure water to prepare 10 mL of a second stock solution at a concentration of 100 µg/ml, which was then vortexed for 1 min. Individual samples were prepared by diluting the second stock of HA to a 2.5 µg/ml solution (corresponding to 1.85 nM of HA, or 6.3 µM of HA disaccharide) containing the desired amount of peptides or KCl. The concentration of peptides was adjusted with 1.795, 1.82, and 0.85 µM peptide solutions for R9, K9, and G9. For samples containing KCl, the ionic strength of the solutions was adjusted with a 1 mM solution of KCl. For all samples, corresponding references containing the same amount of peptides or the same concentration of KCl, but without HA, were prepared. For the characterization of the ionic strength and pH of the prepared samples, conductivity and pH measurements were performed using a conductivity electrode (HI 76312, Hanna Instruments) and a pH electrode (HI 1330, Hanna Instruments), connected to a benchtop meter (HI 55221, Hanna Instruments) and calibrated with suitable buffer solutions.

**AR-SHS measurements.** Second-harmonic scattering measurements were performed on the same AR-SHS setup as described in refs. 59,60. In our AR-SHS setup, the 1032 nm fundamental beam is generated by a mode-locked Yb:KGW laser (Pharos-SP, Light Conversion) with a 190 fs pulse duration and a 200 kHz repetition rate. The polarization of the fundamental beam is controlled by a Glan-Taylor polarizer (GT10-B, Thorlabs) and a zero-order half-wave plate (WPH05M-1030). The beam is further filtered using a long-pass filter (FEL0750, Thorlabs) and then focused into the cylindrical glass cuvette containing the sample (LS instruments, 4.2 mm inner diameter) with a plano-convex

lens ($f$ = 7.5 cm). The beam power at the sample was set to 64 mW, corresponding to a fluence at the focus of ~3.4 mJ/cm². The 516 nm SH signal scattered from the water oriented near the polymer chains was collected and collimated with a planoconvex lens ($f$ = 5 cm), polarization-analyzed by a Glan Taylor polarizer (GT10-A, Thorlabs) and filtered by a 516 ± 10 nm filter (CT516/10bp, Chroma) before being focused into a gated photomultiplier tube (H7422P-40, Hamamatsu). The acceptance angle was set to 3.4° for scattering patterns. The signal was acquired with a gated photon counter (SR400, Stanford Research Instruments) with an acquisition time of 1.5 s/$\theta$. Patterns were obtained in steps of 5° from $\theta = -90°$ to $\theta = 90°$ with 0° being the forward direction of the fundamental beam. 20 periods per angle were recorded from a single sample, each corresponding to a 1.5 s integration at a 200 kHz laser repetition rate. Thus, the signal at each angular point represents an average over $20 \times 300{,}000$ laser shots. Two polarization combinations (PPP and PSS) were measured for each sample and are collectively referred to as PXX. The first letter corresponds to the polarization of the outgoing second harmonic beam, while the last two letters correspond to the polarization of the incoming excitation beam (P = parallel and S = perpendicular to the scattering plane).

To correct for the SH emission from the solvent phase, both the SH response from the sample solution $I_{PXX,sample}(\theta)$ and the SH response from a reference solution $I_{PXX,solution}(\theta)$ of identical ionic strength but without the sample of interest need to be collected. The reference response is subtracted from the AR-SHS signal of the sample and the obtained difference is then normalized to the isotropic SSS signal of pure water to correct for day-to-day differences in the beam profile. The normalized signal $S(\theta)_{PXX}$ is then given by:

$$S(\theta)_{PXX} = \frac{I_{PXX,sample}(\theta) - I_{PXX,solution}(\theta)}{I_{SSS,H_2O}(\theta)} = \frac{I_{PXX}}{I_{SSS,H_2O}(\theta)} \quad (1)$$

### Multi-angle dynamic light scattering

For each sample prepared for AR-SHS measurements, the distribution of the effective size of the solute was measured by multi-angle dynamic light scattering (DLS, Zetasizer Ultra, Malvern). In multi-angle DLS, the correlation data from several detection angles (13°, 90°, and 174°) are combined with the knowledge of Mie theory to generate a higher resolution particle size distribution. Because fitting across multiple representations of the same sample decreases the experimental noise, a more reliable distribution can be calculated, which is particularly useful in polydisperse samples. Despite this improvement with respect to single-angle DLS, for samples of pure HA, HA with KCl, and HA with G9, at least one of the angles could not be collected, resulting in a failed measurement. For all other samples, the Z-average parameter obtained from the intensity-weighted mean hydrodynamic size in the backscattering angle (174°) is listed as the effective size in Table 1, to specifically identify the formation of submicrometer structures and reduce the influence of larger-sized populations, which rather appear at forward scattering angles.

### Microscopy imaging

Stock solutions of 1375 kDa HA, R9, and K9 were prepared by dissolving the samples in water and adjusting them to final concentrations of 13.7 mM for HA disaccharides and 15.75 mM for peptide amino acids. Equal volumes of the HA and peptide solutions were mixed, resulting in a disaccharide-to-amino acid ratio of 0.87. This ratio is the same as the one used in the AR-SHS experiments. After gentle stirring, visible white molecular clusters formed in the mixture. Additionally, experiments with the same composition and 0.5 M NaCl were performed. In the case of G9, a solution of 15.75 mM was mixed with an equal volume of the previously prepared HA solution. Additional experiments with 0.2 M and 0.5 M NaCl were also performed. The pH ranged from 5 to 7 in all cases. All images were captured using an Olympus CKX53 microscope with ×10 magnification. The images were collected with an Olympus EP50 camera attached to the microscope. The

field of view of the microscope setup is 0.46 mm × 0.26 mm. For each sample preparation, one replicate was measured.

## Nuclear magnetic resonance

**Sample preparation.** The deuterium oxide solutions of 1375 kDa HA, R9, and K9 were prepared at the same initial concentrations as for microscopy imaging. However, due to the formation of precipitates, straightforward use of NMR was not possible. Instead, to investigate the thermodynamics of HA–peptide interactions, we collected the $^1H$ NMR spectra of the supernatants from samples prepared at varying HA dimer to amino acid ratios—1:1, 2:1, and 4:1 volume ratios from the stock solutions, corresponding to concentration ratios of 0.87:1, 1.74:1, and 3.48:1. These mixtures were centrifuged for 1 minute, after which the pellets were discarded and the collected supernatants were used for $^1H$ NMR analysis.

Mixtures of smaller 8–15 kDa HA and R9/K9 at higher peptide concentrations were prepared to collect NOE signals. The HA and peptides were each dissolved in deuterium oxide to concentrations of 22 mM, 26.7 mM, and 4.4 mM for R9 amino acid, K9 amino acid, and HA disaccharides, respectively. Equal volumes of these solutions were then combined to achieve amino acid-to-HA dimer ratios of 5:1 and 6:1 for R9 and K9 mixtures, respectively. Slight clouding was observed in the mixtures, but no visible formation of molecular clusters occurred. No intermolecular NOE signals were detected.

**Spectra acquisition.** The $^1H$ spectra of all solutions were collected at 400 MHz Bruker AVANCE III spectrometer ($^1H$ at 401 MHz) equipped with a liquid-nitrogen cryoprobe. In the experiments with different concentration ratios, peptide concentration was determined by integrating a clearly identifiable peptide $^1H$ signal—located between 3.3–3.1 ppm for R9 and 3.1–2.9 ppm for K9—while for following the disappearance of HA we used the peak at 2.2–1.8 ppm, corresponding to the acetamide group. NOESY spectra were recorded on a Bruker AVANCE III 600 MHz spectrometer ($^1H$ at 600.1 MHz) and/or Bruker AVANCE III 500 MHz spectrometer ($^1H$ at 500.0 MHz). Every spectrum was obtained at ~ 295 K. For each sample preparation, one replicate was measured. All spectra were referenced to the solvent signal $\delta(^1H)$ = 4.79 ppm ($D_2O$). Peak assignment followed our previous work[52] as the samples share similar structures (R4, K4, and HA octamer), differing only in size and the presence of protecting groups on the peptides. The resulting spectra were analyzed using Mnova software[70].

## All-atom molecular dynamics simulations

**Simulation models.** All simulated systems are summarized in Table 2. The simulation topologies for the hyaluronan octadecamer (HA18) and the R9, K9, and G9 peptides were prepared using the CHARMM-GUI online utility[71]. The initial zwitterionic conformations for the peptides were prepared in PyMOL[72], while the starting structure for the HA molecules was directly generated in CHARMM-GUI.

Two types of setups were simulated: (1) pure solutions containing either 5 HA chains or 5 peptides, and (2) mixed solutions containing 5 HA chains and 5 peptides of a specific type. The mixed solutions have a 1:1 ratio of amino acids / HA disaccharides, very close to the 0.87:1 ratio used throughout the experimental section of this work. All systems were prepared by initially placing the solutes in an 11 nm side cubic simulation box, i.e., HA and the peptides were at a concentration of 6.2 mM. The simulation boxes were then solvated with approximately 32,000 TIP3P water molecules[73,74]. To neutralize the system net charge, $Na^+$ or $Cl^-$ counterions were added when necessary.

**Simulation protocol.** The AA-MD simulations were carried out using the CHARMM-based biomolecular prosECCo75 force field[62], which provides parameters for hyaluronan, peptides, and ions (Na_s and Cl_2s for sodium and chloride, respectively, see https://gitlab.com/sparkly/

**Table 2 | Simulated systems, their compositions, and corresponding simulation times**

| Simulated systems | # of replicas and simulation times |
|---|---|
| 5 × HA18 + 5 × R9 | 5 replicas: 5 μs, 4.8 μs, 2 μs, 2 μs, 2 μs |
| 5 × HA18 + 5 × K9 | 5 replicas: 2 μs, 2 μs, 2 μs, 2 μs, 1.9 μs |
| 5 × HA18 + 5 × G9 | 5 replicas: 2 μs, 2 μs, 2 μs, 2 μs, 1 μs |
| 5 × HA18 | 5 replicas: 2 μs, 2 μs, 2 μs, 1.7 μs, 1.7 μs |
| 5 × R9 | 2 replicas: 2 μs, 1 μs |
| 5 × K9 | 2 replicas: 2 μs, 1 μs |
| 5 × G9 | 2 replicas: 2 μs, 1 μs |

The total simulated time is ~53 μs. All the simulations and files needed to replicate them can be found on Zenodo https://doi.org/10.5281/zenodo.15115660.

prosecco/prosECCo75). All simulations were performed in the GROMACS simulation package[75], versions 2022.3 and 2023.1. Buffered Verlet lists were utilized to manage atomic neighbors. Electrostatic interactions had a cutoff of 1.2 nm, with the long-range contributions calculated using the smooth particle mesh Ewald (PME) method[76]. The cutoff for Lennard-Jones interactions was set to 1.2 nm, with forces smoothly switched to zero between 1.0 and 1.2 nm[77]. The system temperature was kept at 310.15 K using the v-rescale thermostat with a time constant of 1 ps[78]. Pressure was maintained at 1 bar using the c-rescale barostat, with a time constant of 5 ps and a compressibility of $4.5 \times 10^{-5}$ bar$^{-1}$ [79]. The SETTLE algorithm[80] was employed to constrain water geometry, while the LINCS algorithm[81] was used to constrain all other covalent bonds involving hydrogens.

Each system initially underwent energy minimization, followed by a brief equilibration in the *NPT* ensemble. After equilibration, the systems were propagated for a production run of at least 1 μs, with a 2 fs integration time step. To ensure the convergence and reproducibility of results, two and five replicas were simulated for single-component systems and mixtures, respectively, Table 2. This was done by randomly assigning initial velocities in each replica following the Maxwell–Boltzmann distribution at 310.15 K. For analysis, the first 100 ns of the production simulations, which served as equilibration, were discarded. The trajectories were analyzed using custom scripts written in Fortran or Python, with the Python scripts utilizing the MDAnalysis library[82]. In all cases, the data shown represent the average values across all replicas, with standard deviations included when applicable.

For the autocorrelation analysis, the minimum distance between every residue pair was included in the time series, following a similar approach to Ivanović and Best[83]. This rendered 90 × 90 and 90 × 45 matrices for the HA–HA and HA–peptide distance timeseries, respectively. For the HA–HA autocorrelations, the distances from a HA chain to itself were removed before computation.

## Reporting summary

Further information on research design is available in the Nature Portfolio Reporting Summary linked to this article.

## Data availability

The data and materials associated with this work are available on Zenodo under https://doi.org/10.5281/zenodo.15115544 (a collection of microscope images and videos), https://doi.org/10.5281/zenodo.15115660 (all the simulations and files needed to replicate them), and https://doi.org/10.5281/zenodo.17122576 (NMR data and spectra). All raw data used for the plots in the main text, including AR-SHS data, are available on Zenodo under https://doi.org/10.5281/zenodo.17611850.

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

## Acknowledgements

M.R.-F. acknowledges the support from the Charles University in Prague and the International Max Planck Research School in Dresden. B.C. and

A.M. acknowledge the support of the Julia Jacobi Foundation, as well as the support from the Marie Skłodowska-Curie Actions Innovative Training Network (H2020-MSCA-ITN-2019, proposal 860592, PROTON) provided to B.C. A.K. acknowledges the financial support from Gdańsk University of Technology by the DEC-07/2022/IDUB/II.1/AMERICIUM grant under the AMERICIUM - 'Excellence Initiative - Research University' program. We gratefully acknowledge Poland's high-performance Infrastructure PLGrid (HPC Centers: ACK Cyfronet AGH, PCSS, CI TASK, WCSS) for providing computer facilities and support within computational grant no. PLG/2023/016528. Computations were carried out using the computers of Center of Informatics Tricity Academic Supercomputer & Network. We acknowledge VSB – Technical University of Ostrava, IT4Innovations National Super-computing Center, Czech Republic, for awarding this project access to the LUMI supercomputer, owned by the EuroHPC Joint Undertaking, hosted by CSC (Finland) and the LUMI consortium through the Ministry of Education, Youth and Sports of the Czech Republic through the e-INFRA CZ (grant ID: 90254), project OPEN-32-31. The authors acknowledge the use of Grammarly and ChatGPT for assistance in improving the readability and language of the manuscript. The authors remain solely responsible for the scientific content and interpretations presented.

## Author contributions

A.M., D.B., and H.M.-S. designed and conceptualized the content of this work. M.R.-F. and A.K. carried out the molecular dynamics simulations and data analysis under the supervision of D.B. and H.M.-S. M.R.-F. carried out the NMR and Microscopy experiments and analyzed the data under the supervision of H.M.-S. B.C. carried out the SHS and DLS experiments and analyzed the data under the supervision of A.M. M.R.-F., D.B., and H.M.-S. wrote the manuscript. A.M. carried out a major revision of the manuscript. B.C. and A.K. contributed to the writing of the manuscript. M.R.-F., A.K., and B.C. contributed equally to this work and are allowed to change the publication order to list themselves as first in their CVs.

## Competing interests

The authors declare no competing interests.
