## [Transparent Peer Review file · Communications Chemistry]

Transient formation of supramolecular complexes between hyaluronan and oligopeptides at submicromolar concentration

Corresponding Author: Dr Hector Martinez-Seara

Version 0:

Reviewer comments:

Reviewer #1

(Remarks to the Author)

The manuscript by Hector Martinez-Seara and co-workers reports the use of unconventional techniques to study the supramolecular complexes between the negatively charged hyaluronan (HA) polymer and cationic peptides (nonarginine and nonlysine, R9 and K9) at submicromolar range and using nonaglycine (G9) as neutral control peptide. They propose angle-resolved femtosecond elastic second-harmonic scattering (AR-SHS) in conjunction with all-atom molecular dynamics (AA-MD) as a powerful toolkit for assessing polymer-rich systems, an area currently poorly understood. Two additional techniques, DLS and NMR, were also conducted as part of this integrated approach. The authors claim the outcomes of such analyses could open new possibilities for future applications in material science and biological processes.

The authors have previously published a paper examining the interactions between HA and polyarginine peptide using NMR and AA-MD simulations. This paper is a follow-up study using alternative techniques (e.g. AR-SHS) sensitive to submicromolar concentrations and additional peptides to gain further insights on how HA-peptide complexes regulate the water environment which is considered novel.

Considering the biomedical application of HA, the study has significance for developing supramolecular HA-peptide biomaterials which have been gaining increasing interest. By providing a different angle into the HA-peptide complexes, it provides novel insights for better designing such type of biomaterials, as well as studying other complexes formed by water-soluble (macro)molecules. To enhance the readiness of the manuscript, especially for those working in the field, and make it more impactful, the literature in this area should be better covered in the manuscript, including more pertinent examples on the use of cationic peptides for supramolecular complexation with HA. Also, as some of the techniques used in the study are less familiar to the common reader, such as angle-resolved second harmonic scattering (AR-SHS), a brief introduction to the technique would be beneficial to larger audiences.

The manuscript provides novel insights on the structural organization of ionic complexes. However, there are several less clear aspects in the manuscript that require further clarification, as well as the inclusion of missing information, as outlined below:

- 1) Peptide design: nonapeptides containing arginine (R) and lysine (K) residues were proposed to study interactions with HA. The rational design was based by the presence of Arg residues in HA-binding proteins. While HA-binding proteins bind to HA through domains containing clusters of Arg, these domains are not fully charged and contain other amino acids. It is not clear the rational for using peptides containing fully charged residues. The inclusion of fully charged residues might be responsible for the transient supramolecular complexes formed between HA and the peptides.
- 2) Peptide synthesis: peptides (R9, K9 and G9) were synthesised in the lab. While their synthesis by solid phase was briefly described, the functionalities of the N- and C-terminus of the peptides are not provided. Was the N-terminus acetylated, or is it a free amine? This information is important to assess the full charge of the peptides, in particular for G9. The characterization of the peptides is not provided (MS, HPLC) to ensure their correct identity and purity. Inclusion and discussion of these data (MS and HPLC traces) are necessary to ensure quality control of the synthesised peptides.
- 3) Was the pH of peptide solutions checked and/or adjusted?
- 4) Microscopy images were taken of the HA-peptide mixtures, but these images were not very informative. Was scanning or

transmission electron microscopy (SEM, TEM) attempted? EM might give more details on the nanostructural organization of the complexes, unless their instability prevents further processing of the samples for SEM or TEM observation. Figure S5 shows images of HA-G9 mixture which does not show a translucent solution as described in the text, unless the image provided is not clear. The apparent formation of HA aggregation with G9 requires more clarification/discussion.

5) Figure S7: requires some clarity improvement because the same symbols are used to display different data.

6) Statistical analysis requires more details, especially when there is a graphical display of data (e.g. Figure 2, Figure S6-S8), such as the number of replicates. Data in Table S1 is given without standard deviations. Were the data obtained from single measurements?

Reviewer #2

(Remarks to the Author)

I co-reviewed this manuscript with one of the reviewers who provided the listed reports. This is part of the Communications Chemistry initiative to facilitate training in peer review and to provide appropriate recognition for Early Career Researchers who co-review manuscripts.

Reviewer #3

(Remarks to the Author)

In this manuscript, the authors investigated the interactions between the biopolymer hyaluronan (HA) and various oligopeptides—nonaarginine (R9), nonalysine (K9), and nonaglycine (G9). They accomplished this by integrating advanced techniques like angle-resolved second harmonic scattering (AR-SHS) with conventional methods and molecular dynamics simulations. The study revealed that HA selectively binds to the cationic peptides K9 and R9, with the arginine-rich R9 forming nanoscale clusters. This binding interaction is claimed to restructure the surrounding solvent and solutes. The combined experimental and computational approach provides detailed, atomic-level insight into these transient, multivalent interactions, establishing a new methodology for understanding dynamic polymer systems in soft materials, especially when exploring submicromolar concentrations. This manuscript is suitable for publication in Nature Communications Chemistry, but it requires a major revision addressing the following comments:

1. Figure 1 shows macroscopic optical images of aggregates formed by HA with K9 and R9. The observed differences in aggregate morphology are given as the reason for the different binding behaviors of K9 and R9 to HA. It is relatively difficult to draw a direct correlation between binding and morphology, so this section of the manuscript should be restructured.
2. All tested oligopeptides have charged end groups. The potential influence of these end groups should be discussed in the manuscript.
3. On page 5, the term "state-of-the-art AA-MD" is used. The phrase "state-of-the-art" should be removed.
4. The non-binding behavior with shorter hyaluronan chains is not clearly explained and requires further elaboration.
5. The manuscript is quite long and can be shortened without losing any relevant, important information. Some parts of the main text can therefore be moved to the Supplementary Materials.

Reviewer #4

(Remarks to the Author)

The PDF file containing my comments is attached.

Version 1:

Reviewer comments:

Reviewer #1

(Remarks to the Author)

While most of the questions/comments have been appropriately answered in the rebuttal letter and addressed in the revised manuscript, there are few questions that require further clarification/correction:

- The suggestion by reviewer #4 of including the chemical structures of the peptides and HA in the manuscript is excellent. This was attempted by including them in Figure 1. However, their chemical structure is not fully clear. Please include the chemdraw to better see the differences in the side chains as well as the functionalities present at the N- and C-terminus. Regarding the question about the groups at the termini, it was answered that all the peptides are uncapped with a free amine and a free carboxylate at the N- and C-termini, respectively. However, because a Rink amide MBHA resin was used during synthesis, an amide is expected at the C-terminus and not a free carboxylic acid. Please correct and these functionalities should be clearly displayed in the chemdraw, as well as the expected mass to confirm if the mass observed by MS (Figure S19) corresponds to the peptides. We also realised that the protocol for cleaving the peptides from the resin was not included in the Methods. These details are important for replication of the work and should be added in the manuscript. Purity of peptides is checked/calculated by HPLC and MS, not by NMR.
- The justification given for not using TEM/SEM is not sufficiently convincing.

- Regarding the statistical analysis, 20 measurements were done, but it is unclear whether these were from 20 replicates or 20 analyses of the same sample?
- The last paragraph in the discussion seems somewhat repetitive of what has already been stated throughout the discussion and the manuscript.
- The revised manuscript was submitted without numbers in the Figures (Figure ?). Make sure revision is complete before submitting.

Reviewer #2

(Remarks to the Author)

I co-reviewed this manuscript with one of the reviewers who provided the listed reports. This is part of the Communications Chemistry initiative to facilitate training in peer review and to provide appropriate recognition for Early Career Researchers who co-review manuscripts.

Reviewer #3

(Remarks to the Author)

The author have addressed all my questions.

This study totally suits to the readership of Nature Communication Chemistry journal.

We thank the reviewers for the positive feedback on our work. Below, we provide a point-by-point response to their questions and comments.

Reviewer 1:

The manuscript by Hector Martinez-Seara and co-workers reports the use of unconventional techniques to study the supramolecular complexes between the negatively charged hyaluronan (HA) polymer and cationic peptides (nonaarginine and nonalysine, R9 and K9) at submicromolar range and using nonaglycine (G9) as neutral control peptide. They propose angle-resolved femtosecond elastic second-harmonic scattering (AR-SHS) in conjunction with all-atom molecular dynamics (AA-MD) as a powerful toolkit for assessing polymer-rich systems, an area currently poorly understood. Two additional techniques, DLS and NMR, were also conducted as part of this integrated approach. The authors claim the outcomes of such analyses could open new possibilities for future applications in material science and biological processes.

The authors have previously published a paper examining the interactions between HA and polyarginine peptide using NMR and AA-MD simulations. This paper is a follow-up study using alternative techniques (e.g. AR-SHS) sensitive to submicromolar concentrations and additional peptides to gain further insights on how HA-peptide complexes regulate the water environment which is considered novel.

Considering the biomedical application of HA, the study has significance for developing supramolecular HA-peptide biomaterials which have been gaining increasing interest. By providing a different angle into the HA-peptide complexes, it provides novel insights for better designing such type of biomaterials, as well as studying other complexes formed by water-soluble (macro)molecules.

To enhance the readiness of the manuscript, especially for those working in the field, and make it more impactful, the literature in this area should be better covered in the manuscript, including more pertinent examples on the use of cationic peptides for supramolecular complexation with HA.

Reply: We thank reviewer for the comment. Apart from arginine clusters being present in most HA binding proteins, polyarginines have also been previously used to produce HA-based nanoparticles and control their aggregation [1, 2]. Furthermore, R9 is a known cell-penetrating peptide [3], meaning that its interactions with HA are relevant in the context of R9 penetration mechanism. Many other cell-penetrating compounds have been described as causing glycoalyx aggregation [4], consistent with our work. Following the reviewer’s recommendation, we have extended the contextualization of our setup in the introduction and discussion sections, including additional sources.

Also, as some of the techniques used in the study are less familiar to the common reader, such as angle-resolved second harmonic scattering (AR-SHS), a brief introduction to the technique would be beneficial to larger audiences.

Reply: A brief explanation of AR-SHS and its capabilities was originally included in the Introduction and Methods; we have now moved part of the text from the Methods to the main text to provide a better explanation of the technique just before the presentation of the AR-SHS results.

The manuscript provides novel insights on the structural organization of ionic complexes. We thank the reviewer for the positive evaluation of our work.

However, there are several less clear aspects in the manuscript that require further clarification, as well as the inclusion of missing information, as outlined below:

- **Question/comment 1:** Peptide design: nonapeptides containing arginine (R) and lysine (K) residues were proposed to study interactions with HA. The rational design was based by the presence of Arg residues in HA-binding proteins. While HA-binding proteins bind

to HA through domains containing clusters of Arg, these domains are not fully charged and contain other amino acids. It is not clear the rationale for using peptides containing fully charged residues. The inclusion of fully charged residues might be responsible for the transient supramolecular complexes formed between HA and the peptides.

- **Answer 1:** We have previously shown that sequences containing arginine residues mixed with other amino acids bind HA weaker than pure polyarginines [5]. Therefore, by using the limiting case, a fully charged peptide (i.e., R9 or K9), we are confident that for cases where sequences are not fully charged, the results presented here describing weak and transient interactions will still be relevant. In addition, fully charged peptides have been extensively used in biomaterials. The introduction and discussion have been extended to support this statement.
- **Question/comment 2:** Peptide synthesis: peptides (R9, K9 and G9) were synthesised in the lab. While their synthesis by solid phase was briefly described, the functionalities of the N- and C-terminus of the peptides are not provided. Was the N-terminus acetylated, or is it a free amine? This information is important to assess the full charge of the peptides, in particular for G9. The characterization of the peptides is not provided (MS, HPLC) to ensure their correct identity and purity. Inclusion and discussion of these data (MS and HPLC traces) are necessary to ensure quality control of the synthesised peptides.
- **Answer 2:** All the peptides are uncapped, and it is now clearly described in the methods section. We have also added both the HPLC and MS data for the peptides in the SI, showing their purity (Figures S18, S19). In addition, the high purity of the peptides could already be seen for K9 and R9 in the SI (Figure S3).
- **Question/comment 3:** Was the pH of peptide solutions checked and/or adjusted?
- **Answer 3:** To avoid potential undesirable interactions of our compounds with a buffer, the pH was not adjusted. We routinely measured the pH with indicator paper for the microscopy images and confirmed that it was between 5 and 7, which was our target range. This range is not only biologically relevant but also ensures that all molecules are in the protonation state of interest (positively charged for R9 and K9, and negatively charged for HA). This is indicated in the Methods section. For the SHS, which is very sensitive to conductivity and therefore pH, and the corresponding DLS experiments, the pH was measured, and it is reported in Table S1.
- **Question/comment 4:** Microscopy images were taken of the HA-peptide mixtures, but these images were not very informative. Was scanning or transmission electron microscopy (SEM, TEM) attempted? EM might give more details on the nanostructural organization of the complexes, unless their instability prevents further processing of the samples for SEM or TEM observation. Figure S5 shows images of HA-G9 mixture which does not show a translucent solution as described in the text, unless the image provided is not clear. The apparent formation of HA aggregation with G9 requires more clarification/discussion.
- **Answer 4:** We thank the reviewer for pointing this out. Our intention with the microscopy images was to show the obvious microheterogeneity of the samples that prevents us from performing NMR. We have clarified this aspect in the text. We do not aim to correlate the microscopy images with the interaction strength or binding mode. Nevertheless, we report

apparent differences between R9, K9, and a clearly distinct G9. Therefore, we have not tried any finer imaging technique such as TEM or SEM. Regarding G9, despite being clearly different from R9 and K9 mixtures, we agree that it requires further clarification on why there is some turbidity in the samples. We believe that the usage of non-capped G9, with a terminal amine, is responsible for such inhomogeneities. To test such a hypothesis, we have incrementally increased the ionic strength of the solution by adding NaCl. In the new Figure S5 in the SI, we see the inhomogeneities diminish at 0.2 M concentration (i.e., \sim biological ionic strength) and essentially disappear at 0.5 M. This indicates that the terminal group electrostatics causes the aggregation, since it can be reversed by increasing the ionic strength. The aggregation also disappears in the cases of R9 and K9, highlighting the already mentioned importance of electrostatics in the aggregate formation. The main text has been adapted accordingly to include this new information.

- **Question/comment 5:** Figure S7: requires some clarity improvement because the same symbols are used to display different data.
- **Answer 5:** The lines were supposed to be of different thickness, but the resulting image did not show enough contrast. We have now improved the figure accordingly. We have also reordered the figures in the SI to follow the order of the main manuscript better. This figure is now S6.
- **Question/comment 6:** Statistical analysis requires more details, especially when there is a graphical display of data (e.g., Figure 2, Figure S6-S8), such as the number of replicates. Data in Table S1 is given without standard deviations. Were the data obtained from single measurements?
- **Answer 6:** The error bars for the AR-SHS patterns are displayed as a representative example on the HA and HA-R9 patterns. These error bars represent the error propagation calculated for the normalized patterns based on the standard deviation of 20 measurements. This information is now included in the main text and in the SI. The pH and conductivity measurements are given as a single measurement simply to verify that there are no impurities in the solution. Longer-lasting measurements may change the pH and conductivity values by dissolution of ambient carbon dioxide in the solution.

Reviewer 2: I co-reviewed this manuscript with one of the reviewers who provided the listed reports. This is part of the Communications Chemistry initiative to facilitate training in peer review and to provide appropriate recognition for Early Career Researchers who co-review manuscripts.

We thank the reviewer for the time spent helping us improve the manuscript.

Reviewer 3: In this manuscript, the authors investigated the interactions between the biopolymer hyaluronan (HA) and various oligopeptides—nonaarginine (R9), nonalysine (K9), and nonaglycine (G9). They accomplished this by integrating advanced techniques like angle-resolved second harmonic scattering (AR-SHS) with conventional methods and molecular dynamics simulations. The study revealed that HA selectively binds to the cationic peptides K9 and R9, with the arginine-rich R9 forming nanoscale clusters. This binding interaction is claimed to restructure the surrounding

solvent and solutes. The combined experimental and computational approach provides detailed, atomic-level insight into these transient, multivalent interactions, establishing a new methodology for understanding dynamic polymer systems in soft materials, especially when exploring submicromolar concentrations. This manuscript is suitable for publication in Nature Communications Chemistry, but it requires a major revision addressing the following comments:

We thank the reviewer for the positive evaluation of our work. Below, we provide point-by-point answers to the comments.

- **Question/comment 1:** Figure 1 shows macroscopic optical images of aggregates formed by HA with K9 and R9. The observed differences in aggregate morphology are given as the reason for the different binding behaviors of K9 and R9 to HA. It is relatively difficult to draw a direct correlation between binding and morphology, so this section of the manuscript should be restructured.
- **Answer:** We would like to clarify that our intention was not to use the microscopic images as evidence for differences in binding behavior. We aimed to point out that, in addition to the binding behavior observed with other techniques, we also noted a morphological difference, which prevented us from using NMR as a main working technique. We acknowledge that the original sentence suggesting a potential link between the structural differences of R9 and K9 and their binding strength may have been misleading. To avoid confusion, we have removed this statement from the revised manuscript.
- **Question/comment 2:** All tested oligopeptides have charged end groups. The potential influence of these end groups should be discussed in the manuscript.
- **Answer 2:** We thank the reviewer for this comment. Indeed, all the peptides have charged end groups, which might influence the interaction with HA. In this regard, we have modified Figure S5, which now details the influence of NaCl on the HA aggregation caused by G9, R9 and K9. Given that in G9, the only charges that the presence of NaCl could screen come from the termini, it is clear that these groups influence binding to some extent. We have added such information to the manuscript.
- **Question/comment 3:** On page 5, the term "state-of-the-art AA-MD" is used. The phrase "state-of-the-art" should be removed.
- **Answer 3:** We have removed "state-of-the-art" as requested.
- **Question/comment 4:** The non-binding behavior with shorter hyaluronan chains is not clearly explained and requires further elaboration.
- **Answer 4:** We thank the reviewer for the comment. The answer lies in the multivalent interactions enhanced by the HA length. We have changed the text in the results and discussion section to clarify the issue.
- **Question/comment:** 5. The manuscript is quite long and can be shortened without losing any relevant, important information. Some parts of the main text can therefore be moved to the Supplementary Materials.

- **Answer:** We understand the reviewer’s concern. As one reviewer specifically requested the inclusion of *additional information* in the main text, while others emphasized the need for clearer contextualization of our work and more detailed atomistic insights, the overall length of the manuscript has not been reduced, although some parts have been shortened. We believe this outcome is a fair response to the collective feedback provided by all reviewers.

Reviewer 4: Riopredre-Fernandez et al. present a study combining experimental approaches—including NMR spectroscopy, dynamic light scattering, and angle-resolved second harmonic scattering—with all-atom molecular dynamics (MD) simulations to investigate the interactions of hyaluronic acid (HA) with three oligopeptides: nonaglutine (G9), nonalcyne (K9), and nonaarginine (R9). The study reveals peptide-dependent solute and solvent restructuring effects, which are interesting in their own right. However, the manuscript currently lacks a deeper mechanistic interpretation based on the atomistic insights available from the MD simulations, and does not address the thermodynamic driving forces underlying the observed behavior.

Below are my specific comments:

We thank the reviewer for the positive evaluation of our work.

- **Question/comment 1:** In Figures 3C, S11, and S12, the authors report the timescales of HA–HA and HA–peptide distances obtained from single-exponential fits to the autocorrelation functions. It is unclear which distance time series were used to compute these autocorrelations—were they calculated from the center-of-mass distances between molecules, or do they include all intermolecular atomic pairs?
- **Answer 1:** We agree with the reviewer that this was not clear in the manuscript. In accordance, we have added the following sentences in the Methods section: “For the autocorrelation analysis, the minimum distance between every residue pair was included in the time series, following a similar approach to Ivanović and Best [6]. This rendered 90×90 and 90×45 matrices for the HA–HA and HA–peptide distance timeseries, respectively.” Similar clarification was also included in the caption of Figures 3C, S11, and S12.
- **Question/comment 2:** The autocorrelation data indicate that the decay time for R9 is significantly longer than for K9. Could the authors specify which particular interactions are responsible for this difference?
- **Answer 2:** We have added the following sentence to the discussion, which refers to our previous work and answers the reviewer’s concerns: “The planar guanidinium side chain of R9 allows it to engage in non-polar interactions with the HA chains, in contrast with the purely electrostatic interaction of K9 [5]. This additional non-polar interaction directly contributes to the higher aggregation and slower dynamics of HA–R9 mixtures.”
- **Question/comment 3:** Water dielectric and desolvation effects often play dominant roles in the complexation of charged molecules in solution. Could the authors comment on how these factors influence HA–HA self-interactions and HA interactions with the charged oligopeptides, and compare these findings with the charge-neutral G9 case?

- **Answer 3:** HA–HA is fully soluble at biologically relevant ionic strengths, producing clear solutions. In simulations of pure HA solutions, direct contacts between HA chains are barely seen, and the same is true when adding 150 mM NaCl. When mixing HA with peptides, ionic strength may play some role in decreasing the HA–peptide interaction by attenuating electrostatic attraction. As explained in Question 4 from the reviewer one, the addition of NaCl can experimentally remove the interactions between HA and G9 that create the turbidity when only counterions are present. Regarding desolvation, both Figure S9 (changes in solvent accessible surface area (SASA) upon binding) and S17 (localization of the water molecules in the mixtures) address this issue. From the change in SASA, it is evident that K9 and especially R9 desolvate the complex. The situation is much less noticeable in G9. In addition, Figure S17 shows a clearly different organization of the water molecules in the HA–G9 complex compared to HA–R9 and HA–K9. Yes, ionic strength, dielectric constant, and desolvation all play a role in defining the nature of the reported interactions.
- **Question/comment 4:** Including the chemical structures of the monomeric units of HA and the oligopeptides in one of the figures would greatly aid readers in following the discussion.
- **Answer 4:** We thank reviewer for this comment. We have included the chemical structures of the monomeric units of HA and the oligopeptides in Figure 1

References

- [1] Felipe A. Oyarzun-Ampuero, Francisco M. Goycoolea, Dolores Torres, and Maria J. Alonso. A new drug nanocarrier consisting of polyarginine and hyaluronic acid. *Eur. J. Pharm. Biopharm.*, 79(1):54–57, sep 2011.
- [2] Adam Jugl and Miloslav Pekař. Hyaluronan-arginine interactions-an ultrasound and ITC study. *Polymers (Basel)*., 12(9):2069, sep 2020.
- [3] Christoph Allolio, Aniket Magarkar, Piotr Jurkiewicz, Katarína Baxová, Matti Javanainen, Philip E. Mason, Radek Šachl, Marek Cebecauer, Martin Hof, Dominik Horinek, Veronika Heinz, Reinhard Rachel, Christine M. Ziegler, Adam Schröfel, and Pavel Jungwirth. Arginine-rich cell-penetrating peptides induce membrane multilamellarity and subsequently enter via formation of a fusion pore. *Proc. Natl. Acad. Sci. U. S. A.*, 115(47):11923–11928, 2018.
- [4] André Ziegler and Joachim Seelig. Binding and clustering of glycosaminoglycans: A common property of mono- and multivalent cell-penetrating compounds. *Biophysical Journal*, 94(6):2142–2149, 2008.
- [5] Miguel Riopedre-Fernandez, Denys Biriukov, Martin Dračinský, and Hector Martinez-Seara. Hyaluronan-arginine enhanced and dynamic interaction emerges from distinctive molecular signature due to electrostatics and side-chain specificity. *Carbohydr. Polym.*, 325:121568, feb 2024.
- [6] Miloš T. Ivanović and Robert B. Best. All-atom simulations of biomolecular condensates, 8 2025.

Reviewer 1:

While most of the questions/comments have been appropriately answered in the rebuttal letter and addressed in the revised manuscript, there are a few questions that require further clarification/correction:

We thank the reviewer for the feedback and comments. Below, we provide a point-by-point response.

- **Question/comment 1:** The suggestion by reviewer #4 of including the chemical structures of the peptides and HA in the manuscript is excellent. This was attempted by including them in Figure 1. However, their chemical structure is not fully clear. Please include the chemdraw to better see the differences in the side chains as well as the functionalities present at the N- and C-terminus. Regarding the question about the groups at the termini, it was answered that all the peptides are uncapped with a free amine and a free carboxylate at the N- and C-termini, respectively. However, because a Rink amide MBHA resin was used during synthesis, an amide is expected at the C-terminus and not a free carboxylic acid. Please correct and these functionalities should be clearly displayed in the chemdraw, as well as the expected mass to confirm if the mass observed by MS (Figure S19) corresponds to the peptides. We also realised that the protocol for cleaving the peptides from the resin was not included in the Methods. These details are important for replication of the work and should be added in the manuscript. Purity of peptides is checked/calculated by HPLC and MS, not by NMR.

- **Answer 1:** All chemical structures have now been included in the Supporting Information (now Figure S1). Since the structures are large, we opted to keep the simplified version in the main text and refer the reader to the Supporting Information for full details. The molecular masses of the compounds are also indicated in this new figure.

We sincerely thank the reviewer for noticing the issues in the synthetic section. The previous version of the peptide synthesis method was inadvertently taken from an earlier experimental procedure. We have now thoroughly revised this section to accurately describe the real synthetic process in this work, including the detailed protocol for cleavage of the peptides from the resin.

Finally, our comment referencing the NMR spectra in the SI does not replace the information provided by HPLC and MS and was provided only for completeness.

- **Question/comment 2:** The justification given for not using TEM/SEM is not sufficiently convincing.
- **Answer 2:** The combination of techniques employed here, namely DLS, NMR, AR-SHS, and MD simulations, allows us to characterize the structure and dynamics of HA-peptide interactions at the nanoscale and at low concentrations, which are the focus of this work. While TEM or SEM can provide complementary macroscopic morphological information, our present study focuses on characterizing the nanoscale interactions between HA and peptides. This work does not aim to relate macroscopic morphology to binding mode or strength. Furthermore, acquiring TEM or SEM data is not straightforward as it would require resynthesizing all peptides and coordinating access to appropriate microscopy facilities. The requested imaging techniques may offer complementary insights, but they address questions beyond the objectives of the current work.
- **Question/comment 3:** Regarding the statistical analysis, 20 measurements were done, but it is unclear whether these were from 20 replicates or 20 analyses of the same sample?

- **Answer 3:** Measurements were performed for angles ranging from -90° to $+90^\circ$ in 5° increments. For each angle, 20 periods were recorded, each corresponding to a 1s integration at a 200 kHz laser repetition rate. Thus, the signal at each angular point represents an average over 20×200 kHz laser shots. This information is now added to the text.
- **Question/comment 4:** The last paragraph in the discussion seems somewhat repetitive of what has already been stated throughout the discussion and the manuscript.
- **Answer 4:** The paragraph has been removed from the revised manuscript.
- **Question/comment 5:** The revised manuscript was submitted without numbers in the Figures (Figure ?). Make sure the revision is complete before submitting.
- **Answer 5:** Figure numbers are now correctly displayed.

Reviewer 2:

I co-reviewed this manuscript with one of the reviewers who provided the listed reports. This is part of the Communications Chemistry initiative to facilitate training in peer review and to provide appropriate recognition for Early Career Researchers who co-review manuscripts.
We thank the reviewer for their work.

Reviewer 3:

The author have addressed all my questions.
This study totally suits to the readership of Nature Communication Chemistry journal.
We thank the reviewer for the positive answer.